# Cellular and Molecular Biological Alterations after Photon, Proton, and Carbon Ions Irradiation in Human Chondrosarcoma Cells Linked with High-Quality Physics Data

**DOI:** 10.3390/ijms231911464

**Published:** 2022-09-28

**Authors:** Birgit Lohberger, Sandra Barna, Dietmar Glänzer, Nicole Eck, Sylvia Kerschbaum-Gruber, Katharina Stasny, Andreas Leithner, Dietmar Georg

**Affiliations:** 1Department of Orthopaedics and Trauma, Medical University of Graz, 8036 Graz, Austria; 2Department of Radiation Oncology, Medical University of Vienna, 1090 Vienna, Austria; 3MedAustron Ion Therapy Center, 2700 Wiener Neustadt, Austria

**Keywords:** chondrosarcoma, proton irradiation, carbon ions, DNA repair, cell cycle, particle therapy

## Abstract

Chondrosarcomas are particularly difficult to treat due to their resistance to chemotherapy and radiotherapy. However, particle therapy can enhance local control and patient survival rates. To improve our understanding of the basic cellular radiation response, as a function of dose and linear energy transfer (LET), we developed a novel water phantom-based setup for cell culture experiments and characterized it dosimetrically. In a direct comparison, human chondrosarcoma cell lines were analyzed with regard to their viability, cell proliferation, cell cycle, and DNA repair behavior after irradiation with X-ray, proton, and carbon ions. Our results clearly showed that cell viability and proliferation were inhibited according to the increasing ionization density, i.e., LET, of the irradiation modes. Furthermore, a prominent G_2_/M arrest was shown. Gene expression profiling proved the upregulation of the senescence genes CDKN1A (p21), CDKN2A (p16NK4a), BMI1, and FOXO4 after particle irradiation. Both proton or C-ion irradiation caused a positive regulation of the repair genes ATM, NBN, ATXR, and XPC, and a highly significant increase in XRCC1/2/3, ERCC1, XPC, and PCNA expression, with C-ions appearing to activate DNA repair mechanisms more effectively. The link between the physical data and the cellular responses is an important contribution to the improvement of the treatment system.

## 1. Introduction

Chondrosarcoma is the second-most common primary malignant bone tumor after osteosarcoma and represents a heterogeneous group of locally aggressive and malignant entities. Overall survival and prognosis depend on histological grade and tumor subtype [1]. Worldwide, the overall age-standardized incidence rate is 0.1–0.3 per 100,000 per year [2]. Resistance to chemo- and radiotherapy is a consequence of the underlying phenotype, which includes poor vascularization, slow division rate, and a hyaline cartilage matrix that prevents access to the cells. For this reason, the therapy options are limited and complete surgical resection remains the gold standard for primary or recurrent chondrosarcoma [3,4]. Due to their poor radiosensitivity, high doses are recommended in palliative settings, after incomplete resection or for unresectable tumors in anatomically challenging sites. Due to the inherent therapy resistance of chondrosarcomas to both chemo- and radiotherapy, research for better treatment options for unresectable or metastatic chondrosarcoma is imperative.

Particle therapy (PT) with proton or carbon ions (C-ions) enable the improvement of local control and patients’ survival rates compared to photon beam therapy [5]. The primary rationale for proton radiotherapy is dosimetric ballistics, i.e., the sharp dose increases at a well-defined depth (Bragg peak) and the rapid dose falls off beyond that maximum. Thus, a highly conformal high-dose region can be tailored to cover the target volume with high precision, and to simultaneously reduce normal tissue exposure. Additionally, protons are slightly more biologically effective than photons. However, the current clinical practice of assuming a 10% higher and constant relative biological effectiveness (RBE) for protons compared to photons is controversially discussed [6,7]. C-ions have an elevated RBE and are generally assumed to be most efficient for radiation-sensitive tumors. Although clinically applied [8,9], PT for chondrosarcomas lacks large prospective studies as well as comprehensive pre-clinical research to improve the understanding of the basic cellular radiation response.

In order to create the most physiological environment possible to enable reproducible in vitro studies, a novel water phantom-based setup was developed by our working group. The dosimetric characterization for both particle types has been carried out [10]. This setup allows comprehensive physics characterization of particle beams for pre-clinical research, in accordance with the report of a National Cancer Institute special panel [11] suggesting the linkage between high-quality biology data and high-quality physics data that go beyond the macroscopic parameter absorbed dose D in units of Gray. More specifically, microdosimetric parameters such as the linear energy transfer (LET) can be determined in predefined reproducible positions, as these parameters are linked with radiation response. Reduced uncertainties in physical parameters and complementary microdosimetric aspects will pave the way toward improved PT.

To realize the recommended linkage between physical data and biological data, we used this novel measurement setup for cellular and molecular biological analysis regarding viability and cell proliferation, cell cycle progression, and gene expression profiling with reference to DNA repair behavior in two human chondrosarcoma cell lines after proton and C-ion IR. Photon IR was used as reference irradiation (IR) for all experiments. According to the author’s knowledge, such an in-depth molecular biological analysis and respective dependencies on both absorbed dose and LET have so far not been described in the literature.

## 2. Results

### 2.1. Dosimetric Characterization of the Novel Water Phantom Measurement Set-Up

Using PT, a highly conformal high-dose region can be tailored to cover the target volume with high precision while reducing the burden on normal tissue. Figure 1 underlines the clear difference in dose distribution between photon (X-ray) (Figure 1A), proton (Figure 1B), or C-ion IR (Figure 1C).

Figure 2 shows the dose and LET distribution for protons and C-ions determined in the novel water phantom for in vitro experiments, including a comparison between measured and calculated dose values. For the clinically relevant target size and IR depth, proton energies between 66.5 and 135.6 MeV were required for our in vitro study. For the flask position in the middle of the SOBP, this had no significant influence on the proton-dose-averaged linear energy transfer (LET) value of 2.9 keV/μm (Figure 2A). The positioning uncertainty corresponding to a reproducibility of 0.55% on the dose measurements was 300 μm. For C-ions in general, the LET is higher and the LET distribution is steeper. The respective LET range for C-ions with energies between 170 and 230 MeV/u was between 50 and 150 keV/µm, respectively. However, in the middle of the C-ion SOBP, the influence due to positioning errors was still below 0.5% with an LET of 55.2 keV/μm (Figure 2B). The LET is based on Monte Carlo calculations and was derived directly from the treatment planning system. Reference photon IR was performed in a 200 kV beam, generated by an YXLON unit. The following filtration was used for radiobiological experiments: 3 mm Be + 3 mm Al + 0.5 mm Cu. The cell layer was positioned at 40 cm distance from the beam exit window.

### 2.2. Cell Biological Alterations after Photon, Proton, and C-Ions IR in Human Chondrosarcoma Cells

To demonstrate the cell biological influence of the different types of IR, human chondrosarcoma cells were irradiated with 4 Gy X-ray/proton/C-ions and cell proliferation and viability were measured. The real-time xCELLigence system clearly showed that cell proliferation is inhibited according to the increasing LET (=ionisation density) levels of the IR modes (X-ray < proton < C-ions) (Figure 3A). The same effect can be observed in endpoint measurements of viability after 24–168 h (Figure 3B).

Another important aspect in tumor biology is the impairment of the cell cycle by therapeutic interventions. Flow cytometry analysis was performed to determine the effect of IR on cell cycle distribution of chondrosarcoma cultures when exposed to 4 Gy and 8 Gy X-ray/proton/C-ions. Non-IR cells were measured as controls. All values of four individual experiments each (% of gated cells) and their statistical differences are listed in Table 1 (mean ± SD, *n* = 4). The graphical representations of the G_0_/G_1_, S, and G_2_/M values of both cell lines are shown in stacked bars (Figure 4A). High-dose IR caused a highly significant increase in the number of cells in the G_2_/M phase compared to controls, which was accompanied by a decrease in the number of G_0_/G_1_ and S phase cells, indicating a persisting G_2_/M phase arrest at the 24 h time point. The Cal78 cell line responded with a significantly greater shift in cell cycle phases. Representative flow cytometry measurements of non-IR control cells and irradiated with 8 Gy X-ray/proton/C-ions are presented in Figure 4B. In the context of the altered cell cycle, we analyzed the most important genes of different cell cycle phases using RNA expression profiling 1 h, 24 h, and 72 h after X-ray/proton/C-ions IR. A heatmap plot of RNA sequencing data was presented in a log2-transformed fold-change regarding expression of cell cycle regulation genes alterations after IR (Figure 4C). Particularly prominent, senescence genes such as CDKN1A (p21), CDKN2A (p16NK4a), BMI1, and FOXO4 were upregulated after particle IR. The early DNA replication genes PCNA, MSH2, CDC25A, and CCNE2 were especially increased at the 1 h time point, whereas the late replication genes NPAT and ATXR peaked 24 h after IR. Due to the arrest of the cells in the G_2_/M phase, a general downregulation of the corresponding genes could be observed.

### 2.3. Activation of DNA Repair Mechanisms after IR

To investigate to what extent the different types of IR influence the mechanisms of DNA repair, we isolated RNA without IR (ctrl) and 1 h, 24 h, and 72 h after 4 Gy X-ray/proton/C-ions IR and performed RNA expression profiling. In order to represent the relationship of the most important DNA repair key genes, the protein–protein interactions were analyzed using the STRING database (version 11.5; string consortium 2022; http://www.string-db.org), which includes the experimentally determined connections, the gene neighborhood, co-expression, and protein homologies (Figure 5A). A heatmap plot of RNA sequencing data was presented in a log2-transformed fold-change regarding expression of key player genes of the base excision repair (BER), the mismatch mediated repair (MMR), the nucleotide excision repair (NER), the homology directed repair (HDR), and the nonhomologous end-joining (NHEJ) (Figure 5B). Irrespective of the type of IR, genes of the BER pathway play only a subordinate role and were predominantly reduced. On the other hand, the key genes of the NER pathway ERCC1/2/5 were upregulated after 24 and 72 h in all types of IR. The relevance of the HDR pathway, already known from the literature, is underlined by the positive regulation of the genes ATM, NBN, ATXR, and XPC. For the majority of genes, regulation was more pronounced with PT than with photon therapy.

In order to be able to show the regulation of the most important key genes after treatment with the respective types of IR in more detail, we performed RT-qPCR analysis with RNAs isolated 24 h after 4 Gy X-ray/proton/C-ions IR. The gene expression analysis of the DNA repair genes is presented in Figure 6A. PT with proton or C-ions revealed a highly significant increase in XRCC1/2/3, ERCC1, XPC, and PCNA expression, where C-ions seem to activate the DNA repair mechanisms more effectively. MSH2/6, Rad51, and PARP-1 were slightly inhibited in their mRNA expression or showed no changes. Furthermore, MSH3 and XRCC5/6 showed no significant differences and are, therefore, not shown graphically.

To investigate the ability of IR to affect protein phosphorylation levels, whole-cell lysates of SW-1353 and Cal78 chondrosarcoma cells were extracted 1 h and 24 h after 2, 4, and 8 Gy X-ray/proton/C-ion IR and prepared for Western blot analysis (Figure 6B). With increasing IR intensities, the phosphorylation of p53 increased significantly in both cell lines after 1 h. As this is a very rapid cellular response, a dose-dependent increased phosphorylation level of the phosphoinositide 3 kinase-related ataxia-telangiectasia mutated (ATM) and ATM and RAD3-related (ATR) protein kinases was observed 1 h after IR, whereby the effect was best visible after the C-ion IR. Phosphorylation of the DNA damage marker γH2AX also increased in a dose-dependent manner. One representative blot out of three is shown and β-actin was used as a loading control (mean  ±  SD; *n*  =  3). Full-length blots/gels are presented in Appendix A.

## 3. Discussion

PT with protons and C-ions has several advantages compared to conventional radiation therapy with high-energy photon beams. Both protons and C-ions have improved ballistic features, i.e., they travel a fixed distance in tissue that is related to the accelerating energy where they deposit the bulk of their energy (‘Bragg Peak’) [12]. Protons do not deliver a dose beyond the Bragg Peak, but due to fragmentation, C-ions deliver a small fraction of dose to the normal tissues distal to the Bragg Peak [13]. Compared to high-energy photon beams, C-ions have the additional advantage of an elevated RBE. The higher RBE is linked to the higher LET, i.e., the higher ionization density and track structure of secondary IR. Due to its complexity and costs, C-ion treatments today are mainly limited to radiosensitive tissues and/or radioresistant skull base tumors such as chordoma and chondrosarcoma [14,15]. Moreover, although the side effects and sequelae occurring after conventional radiation therapy have been relatively well described, the cellular mechanisms, especially in mesenchyme-derived cells, are still poorly understood. Studying such mechanisms is even more necessary for C-ion IR, considering the tendency toward hypo-fractionated treatments, which lead to higher doses deposited in healthy tissues [16]. All these effects are taken into account in treatment planning systems via biophysical models. A current limitation is linked to the available data and tumor model systems to refine and revise these biophysical models.

Our novel measurement set-up for biological research enables reproducible and accurate cell positioning. This is of high importance for studies investigating any biological effect in the gradient regions of the SOBP, namely the plateau and fall-off regions. The biological effect of the plateau region is of high clinical interest as it corresponds to the skin and healthy tissue of the patient before reaching the target, causing most adverse effects of PT. In the fall-off region, where the dose and LET gradient are steepest, the particles deposit their whole (protons) or most (C-ions) of their energy abruptly over a small distance, making uncertainties caused by organ motion or daily set-up variability higher than for conventional radiotherapy. For this reason, the biological effect of the fall-off region is also of clinical importance, as it has the highest LET and RBE contribution of the whole SOBP. Better understanding of the biological response along an entire SOBP is a substantial step toward better patient outcomes.

Despite the utmost importance for clinical-therapeutic applications, only a few publications on the cellular processes are known. Girard et al. demonstrated that chondrosarcoma cells respond differently to IR due to their strong genetic heterogeneity [17]. In our previous study, we were able to show that both X-ray and proton IR resulted in reduced chondrosarcoma cell survival and a decreased ability to form colonies [18]. In the straight comparison between PT with protons and C-ions, and the reference photon IR, the greatest inhibitory effect on cell viability and proliferation was seen after C-ion IR. Similar to other tumor entities, human chondrosarcoma cells also showed a markedly altered cell cycle and DNA damage repair after protons [19,20] and C-ions [21,22], respectively. In particular, the higher dose of 8 Gy resulted in a transformation in the cell cycle with a decrease in the number of cells in the G_0_/G_1_ phase and S phase, accompanied by a significant increase in the number of G_2_/M phase cells. While the cell cycle behavior was very similar after X-ray and proton IR, a highly significant arrest of the cells in the G_2_/M phase occurred after C-ion IR. However, even the lower dose of 4 Gy was sufficient for highly significant arrest in G_2_/M. Previously, Maity et al. showed that exposing a wide variety of cells to IR resulted in a mitotic delay that involved several events in the G_0_/G_1_, G_2_/M, or S phase, and that the G_2_ arrest was observed in virtually all eukaryotic cells and occurred following high and low doses, even under 1 Gy. The S phase delay was typically seen following higher doses (>5 Gy) [23].

The mechanisms underlying IR-induced G_2_ arrest again shed light on the central role of ataxia-telangiectasia-mutated protein (ATM) in the initiation and maintenance of genomic instability. Being one of the earliest known responders to DNA damage, the ATM signaling cascade is activated within minutes in response to IR, and its protein kinase activity is rapidly enhanced with the ability to phosphorylate its downstream targets involved in DNA repair, cell cycle checkpoint control, and apoptosis processes, which ultimately induces the G_2_ arrest. Our gene expression data showed a significant increase in senescence-associated CDKN1A (p21), CDKN2A (p16NK4a), BMI1, and the forkhead box O (FOXO4) after proton and C-ion IR. Cellular senescence is a permanent arrested state of cell division, induced by various factors including exposure to IR. The senescence process induced by IR starts with DNA damage, after which a G_2_ arrest occurs in virtually all eukaryotic cells and a mitotic bypass is possibly necessary to ultimately establish cellular senescence. Within this complex DNA damage response signaling network, ATM, p53, and CDKN1A (p21) stand out as the crucial mediators [24]. Senescence cells can be identified by prominent β-galactosidase activity, increased p53, CDKN1A (p21), and CDKN2A (p16NK4a) expression, and decreased levels of CDK1 (Cdc2) and survivin (BIRC5) [25]. Exactly this expression pattern was found in irradiated chondrosarcoma cells, whereby the effects are significantly more pronounced after particle IR than after photon IR. In addition, the activation of FOXO4 is correlated with an increased transcriptional activation of CDKN1A (p21) and subsequent activation of cellular senescence [26]. Furthermore, the increase in NPAT expression after proton and C-ion IR is notable, which is required for progression through the G_0_/G_1_ and S phases of the cell cycle, activates transcription of histone genes, and positively regulates the ATM promoter [27].

The complex network of the different DNA repair key players can be represented by means of the STRING database. This illustrates the close interconnection of the main regulators of the individual DNA repair pathways. The maintenance of genomic stability requires a protective cellular response to DNA damaging agents or radiotherapy. This DNA damage response pathway encompasses proteins that detect DNA damage, function in DNA repair pathways, and regulate the cell cycle. Photon IR mainly leads to isolated lesions such as single-strand breaks (SSBs), base damage, and double-strand breaks (DSBs). In contrast, PTs with high LET, such as C-ions, cause more localized and clustered DNA damage [28]. The LET of protons varies along the Bragg curve; hence, the spatial distribution of lesions may be different [29]. The two canonical pathways of DSBs repair are HDR and NHEJ [30], which have been described as the main pathways, with HDR being more relevant after a C-ion IR [31,32]. Our gene expression profiling and RT-qPCR analysis confirmed these findings and revealed increased expression of the HDR genes ATM, ATXR, NBN, and XPC, as well as significant activation of the NER pathway with the regulators ERCC1/2/5 and XPA.

Furthermore, DSBs and SSBs activate a network of post-translational modifications, including phosphorylation [33]. Once activated, ATM and ATR phosphorylate an overlapping pool of substrates to promote DNA repair and coordinate other DNA metabolism processes such as transcription and replication. IR induced rapid protein phosphorylation of p53 in response to DNA damage. ATM operates upstream of p53 in a signal transduction pathway and also showed rapid phosphorylation [34]. The 8 Gy protons and especially C-ions caused a longer persistent phosphorylation in both chondrosarcoma cell lines. The phosphorylation of ATR is clearly more sensitive with Cal78. One of the first events after induction of DSBs by IR is the phosphorylation of γH2AX, which is why the dose-dependent phosphorylation can be seen particularly well 1 h after IR. In this case also, the C-ion IR showed the strongest effect.

For the first time, direct intercomparison demonstrated the link between the radiation physics data and the cellular and molecular biology changes after photon, proton, and C-ion IR in human chondrosarcoma cells. The elucidation of cellular and molecular biological alterations after particle therapy is another important step for the improvement of the treatment regimen of this almost therapy-resistant tumor entity.

## 4. Materials and Methods

### 4.1. Physical Parameters of Irradiation

All IR experiments were performed at MedAustron, the synchrotron-based Austrian center for ion therapy and research. The experimental research room is equipped with a horizontal beam line including an active spot scanning technique with active energy variation for both proton and C-ions. The precise and standardized positioning of IR samples embedded in respective measurement phantoms is facilitated by a high-precision robot couch and a laser positioning system. For the photon reference IR, a dedicated X-ray unit (YXLON Y.TU 320-D03, YXLON GmbH, Hamburg, Germany) was used. The unit was previously commissioned for small animal IR [35] and is equipped with a 3 mm Be/3 mm Al/0.3 mm Cu filter. A current of 20 mA and a voltage of 200 kV were used to achieve a dose rate of 1.3 Gy/min. The cell layer was positioned at a 40 cm distance from the beam exit window. For the proton IR, a treatment plan with a spread-out Bragg peak (SOBP) of 4 cm was designed for a field size of 17 × 9 cm^2^ utilizing the treatment planning system (TPS) RayStation v7.99 (RaySearch Laboratories, Stockholm, Sweden). Dose calculation was performed with a Monte Carlo v4.3 dose engine [36]. To cover the SOBP centered at 8 cm depth, proton energies between 66.5 and 135.6 MeV were required. Similarly, for the C-ion IR, a treatment plan with a spread-out Bragg peak (SOBP) of 4 cm was designed using the same TPS for the same field sizes and depth, requiring C-ion energies between 170 and 230 MeV/u. Ripple filters were used to ensure a flat SOBP. For both particle types, the energy layers were spaced either 1 mm or 2 mm apart. The radiation delivery technique was pencil beam scanning, and no ranger shifters were used to modify the beam.

### 4.2. Dosimetric Verification of the Experimental Setup

Customized holders for radiobiological in vitro experiments in a horizontal beam line were developed and commissioned in-house. Their dosimetric characterization in both proton and C-ion beams was performed using a plane-parallel ionization chamber (Advanced Markus^®^ Electron Chamber (Type 34045, SN 001540)) as well as a cylindrical ionization chamber (0.125 cc Semiflex Chamber (Type 31010, SN 006012)), inserted into modified chamber slide flasks, which were positioned side-by-side for simultaneous IR. Repeated measurements were carried out along the entire depth dose curve, resulting in a reproducibility of 0.55% expressed as one standard deviation. The general positioning uncertainty was assessed by taking into account the reproducibility, resolution of the water phantom scale, and positioning of both ionization chambers [10].

### 4.3. Monte Carlo Simulations

In order to benchmark TPS calculations including dose-averaged LET at the geometric position of the cell layers, Monte Carlo simulations were carried out using the GATE Monte Carlo platform, which is based on the Geant4 toolkit [37]. The treatment plan used for cell IR was converted into a particle source description at the vacuum exit, which was used as input for the validated beam model of our nozzle design [38]. A simple scoring geometry was chosen: a cylinder with a radius of 2.5 mm, corresponding to the collecting electrode dimensions of the Advanced Markus ionization chamber, and a height of 150 mm. The DoseActor and LETActor (both are standard GATE packages) were attached to this scoring cylinder, with a voxel size along the height of the cylinder of 0.75 mm each. The physics list chosen was QGSP_INCLXX_EMZ for protons and Shielding_EMZ for C-ions, and the number of primary particles was 10⁸ over the entire plan.

### 4.4. Cell Culture

SW1353 (primary grade II) (ATCC^®^ HTB-94™, LGC Standards, Wesel, Germany) and Cal78 (recurrence of dedifferentiated grade III) (ACC449; DSMZ, Leibniz, Germany) chondrosarcoma cell lines were cultured in Dulbecco’s modified Eagle’s medium (DMEM-F12) supplemented with 10% FBS, 1% L-glutamine, 1% penicillin/streptomycin, and 0.25 µg amphotericin B (all GIBCO^®^, Invitrogen, Darmstadt, Germany). The cell lines were authenticated by STR profiling within the last three years. All experiments were performed with mycoplasma-free cells. For IR experiments, adherent chondrosarcoma cells in log-growth phase were plated either in a density of 1 × 10^5^ cells/Slideflasks 9 cm^2^ (Thermo Fisher Scientific) or 5 × 10^5^ cells/T25 flasks and incubated overnight at 37 °C with 5% CO_2_.

### 4.5. Viability and Proliferation Analysis

For the dose–response relationship, chondrosarcoma cells were irradiated with 0 Gy (neg. control) and 4 Gy X-ray/proton/C-ions IR. Cell viability was determined with the CellTiter-Glo^®^ cell viability assay (Promega Corporation, Madison, MI, USA) after 24 to 168 h and normalized to the non-IR controls. Background reference values were derived from the culture media. Absorbance was measured with a LUMIstar™ microplate luminometer (BMG Labtech GmbH, Ortenberg, Germany) (mean ± SD; *n* = 7, performed in biological quadruplicates). The xCELLigence RTCA-DP device (OLS, Bremen, Germany) was used to monitor cell proliferation in real-time. Cells were seeded after IR in electronic microtiter plates (E-Plate™, OLS) and measured for 120 h according to the manufacturer’s instructions. Cell density was measured in quadruplicate with a programmed signal detection every 20 min. Data acquisition and analyses were performed with RTCA software (version 1.2, OLS).

### 4.6. Cell Cycle Analysis

A period of 24 h after 0 Gy (control), 4 Gy, and 8 Gy X-ray/proton/C-ions IR, cells were harvested by trypsinization and fixed with 70% ice-cold ethanol. Before flow cytometry analysis, the cell pellet was resuspended in propidium-iodide (PI)-staining buffer (50 μL/mL of PI, 100 µg/mL of RNAse A, 0.1% Natriumcitrat, and 0.1% Triton X-100) and incubated for 20 min at room temperature. Cell cycle distribution was measured with CytoFlexLX (Beckman Coulter, Pasadena, CA, USA) and analyzed using ModFit LT software Version 4.1.7 (Verity software house). Four independent experiments were conducted in each case.

### 4.7. Gene Expression Profiling (RNA Sequencing)

For the next-generation sequencing (NGS), RNA cells were isolated 1 h, 24 h, and 72 h after 0 (control) and 4 Gy X-ray/proton/C-ions IR. Gene expression profiling was performed using the Thermo Fisher Ion Ampliseq RNA workflow. Briefly, RNA was transcribed to cDNA using the SuperScript™ VILO™ cDNA Synthesis Kit according to the manufacturer’s protocol. cDNA equivalent to 50 ng of RNA was used in a PCR reaction with a custom Ion Ampliseq RNA Panel encompassing amplicons in 69 genes. The NGS Library was generated from the PCR product using the AmpliSeq Library Kit Plus and subsequent library quantification was performed using the Ion Library TaqMan™ Quantitation Kit. Sequencing was performed on an Ion S5XL benchtop sequencer using the 540 Chip Kit and the 200 base pairs workflow (all Thermo Fisher Scientific, Waltham, MA, USA) to a total depth of approximately one million reads per sample. Individual gene expression is considered to be equivalent to the relative read count of the gene-specific amplicon in the total library. Data were analyzed using the Ampliseq RNA Ion Torrent Suite Plugin (version 4.4.0.4) and individual gene expression was calculated as amplicon reads per million total reads (RPM). As Ampliseq RNA is an amplicon counting technology, we reported the number of mapped reads, percent reads on target, and percent assigned reads for each sample (*n* = 4).

### 4.8. Quantitative Reverse-Transcription Polymerase Chain Reaction (RT-qPCR)

Total RNA was isolated 24 h after IR with 4 Gy X-ray/proton/C-ions IR using the RNeasy Mini Kit and DNase-I treatment according to the manufacturer’s manual (Qiagen, Hilden, Germany). Two micrograms of RNA were reverse-transcribed with the iScript-cDNA Synthesis Kit (BioRad Laboratories Inc., Hercules, CA, USA) using a blend of oligo(dT) and hexamer random primers. Amplification was performed with the SsoAdvanced Universal SYBR Green Supermix (Bio-Rad Laboratories Inc.) using technical triplicates and measured by the CFX96 Touch (BioRad Laboratories Inc.). The following QuantiTect primer assays (Qiagen) were used for real-time RT-PCR: XRCC1, XRCC2, XRCC3, ECRR1, XPC, MSH2, MSH3, MSH6, PCNA, Rad51, PARP-1, XRCC5, and XRCC6. Results were analyzed using the CFX manager software for CFX Real-Time PCR Instruments (Bio-Rad Laboratories Inc., version 3.1) software and quantification cycle values (C_t_) were exported for statistical analysis. Results with C_t_ values greater than 32 were excluded from analysis. Relative quantification of expression levels was obtained by the ∆∆Ct method based on the geometric mean of the internal controls ribosomal protein, large, P0 (RPL), and TATA box binding protein (TBP). The expression level (C_t_) of the target gene was normalized to the reference genes (ΔC_t_), and the ΔC_t_ of the test sample was normalized to the ΔC_t_ of the control (ΔΔC_t_). Finally, the expression ratio was calculated with the 2^−ΔΔCt^ method.

### 4.9. Protein Expression Analysis

Whole-cell protein extracts were prepared with lysis buffer (50 mM Tris-HCl pH 7.4, 150 mM NaCl, 1 mM NaF, 1 mM EDTA, 1% NP-40, and 1 mM Na3VO4) and a protease inhibitor cocktail (P8340; Sigma Aldrich), 1 h and 24 h after 2, 4, and 8 Gy X-ray/proton/C-ions IR. Protein concentration was determined with the Pierce BCA Protein Assay Kit (Thermo Fisher Scientific). The proteins were separated by SDS-PAGE and were blotted on Amersham™ Protran™ Premium 0.45 µM nitrocellulose membranes (GE Healthcare Life Science, Little Chalfont, UK). Primary antibodies against the DNA damage key players phospho-p53, phospho-ATM, phospho-ATR, the death receptor TRAIL-R2, the DNA damage marker phospho-histone γH2AX (Cell Signaling Technology, Danvers, MA, USA), and β-actin (Abcam, Cambridge, UK) as the loading control were used. Blots were developed using a horseradish peroxidase-conjugated secondary antibody (Dako) for 1 h and the Amersham™ ECL™ prime Western blotting detection reagent (GE Healthcare). Chemiluminescence signals were detected with the ChemiDocTouch Imaging System (BioRad Laboratories Inc., Hercules, CA, USA) and respective images were processed with the ImageLab 5.2 Software (BioRad Laboratories Inc.).

### 4.10. Study Limitation

The results presented above were obtained for two cell lines with protons and C-ions. Adding a third or fourth chondrosarcoma line was not feasible for logistic reasons and the allocated annual research beam time for ion beam research.

Although protons and C-ions represent the currently used particle species utilized clinically, results from a third ion species such as helium with an LET value between protons and C-ions would be optimal to confirm our results and to the close the gap between the high (C-ions) and low LET range (protons). However, helium ion beams are scarce even at the global scale and will not become available at our research facility until 2025.

The novel phantom with a customized holder for cell positioning might introduce a small geometric uncertainty in the sub-millimeter range. In this study, all cells were irradiated in the middle of a SOBP having a width of 4 cm. Even if we assume an excessive positioning uncertainty of 1 mm, the proton LET would still not change significantly, while the C-ion LET would change by a maximum of 1 keV/µm in this constant-dose region. These geometric uncertainties and the associated LET effects are negligible within the scope of our study.

## Figures and Tables

**Figure 1 ijms-23-11464-f001:**
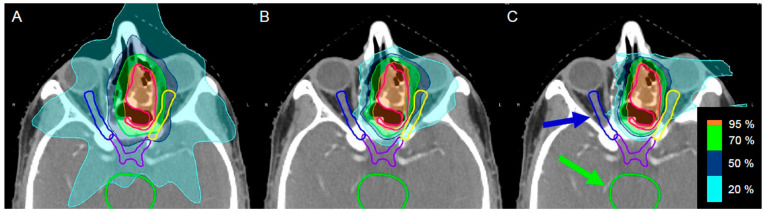
Illustration of the different treatment techniques, shown on a representative booster plan for the gross tumor volume (red) in the paranasal sinus region. The conventional treatment method ((**A**) volumetric modulated arc therapy) shows a higher dose delivered to organs at risk, e.g., the chiasm (purple), both optic nerves (yellow and blue), and the brainstem (green). The dose distribution of the pencil beam scanning technique with (**B**) protons and (**C**) C-ions show better sparing of the organs at risk, especially visible in the right optical nerve and brainstem, while achieving a comparable tumor coverage.

**Figure 2 ijms-23-11464-f002:**
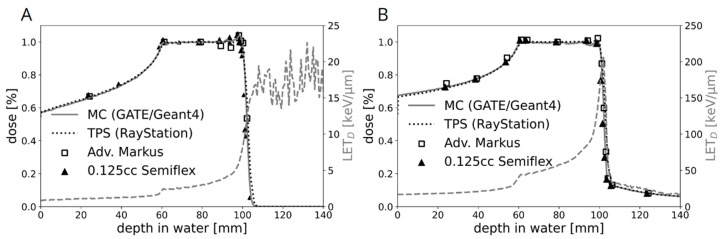
The percentage depth dose curves of the (**A**) proton and (**B**) C-ion SOBP used for cell irradiations. A very good agreement between the predicted dose distributions of the treatment planning system (TPS) and Monte Carlo (MC) simulation was achieved. The respective LET spectrum shows the increase with increasing depth in water. C-ions have a much higher LET than protons, which is reflected by the scale of the mirrored y-axis (25 keV/μm versus 250 keV/μm). For protons, the LET after the fall-off region has no physical meaning as no particles are delivered beyond this depth.

**Figure 3 ijms-23-11464-f003:**
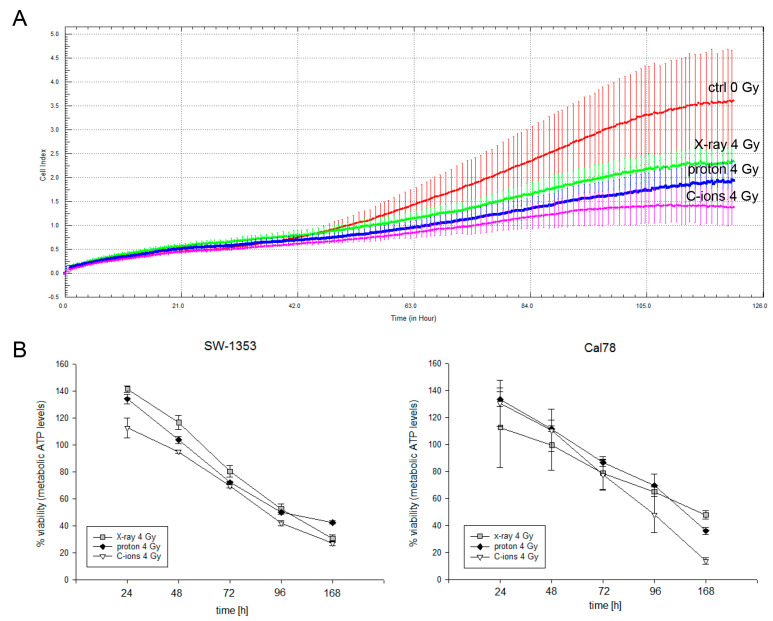
Proliferation and cell viability analysis after 4 Gy X-ray/proton/C-ions IR. (**A**) xCELLigence real-time proliferation analysis (red: non-IR controls; green: X-ray 4 Gy; blue: proton 4 Gy (LET 2.9 keV/μm); magenta: C-ions 4 Gy (LET ca. 55 keV/μm)); (**B**) the percentage of metabolic ATP levels, which is representative for the viability of the cells (mean ± SD; *n* = 3; measured in quadruplicates). The direct comparison of the three types of IR showed a slight gradation of the cellular response with the strongest effect with C-ions.

**Figure 4 ijms-23-11464-f004:**
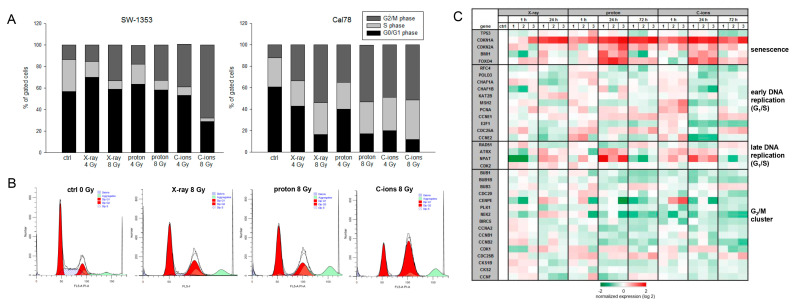
Cell cycle analysis. (**A**) Chondrosarcoma cells were analyzed using flow cytometry 24 h after 4 Gy and 8 Gy X-ray/proton/C-ions IR. The corresponding statistical evaluation shown in stacked bar charts (*n* = 4). (**B**) Representative original tracks of non-IR control cells (ctrl) and one measurement of each IR type (8 Gy) are shown. (**C**) Heatmap blot of RNA sequencing data of relevant cell cycle regulator genes presented in log2 fold-change 1 h, 24 h, and 72 h after 4 Gy X-ray/proton/C-ions IR (*n* = 4).

**Figure 5 ijms-23-11464-f005:**
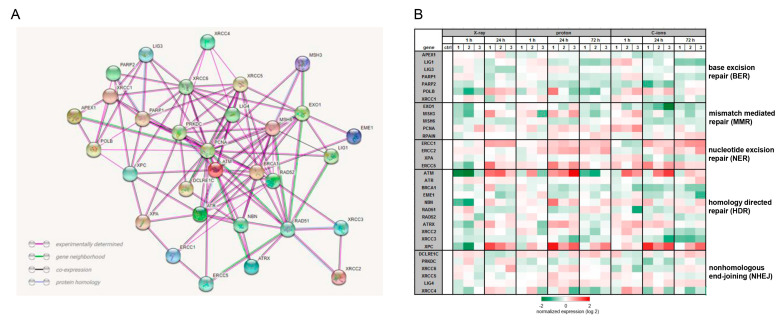
Expression of DNA repair key genes. (**A**) The interaction network was prepared using the STRING database (version 11.5; string consortium 2022; http://www.string-db.org), collected from different databases. Experimental determined connections were presented with magenta lines, gene neighborhood (green), co-expression (black), and protein homologies (blue). (**B**) Heatmap blot of RNA next generation sequencing data of the key players of DNA repair mechanism pathways presented in log2 fold change 1 h, 24 h, and 72 h after 4 Gy X-ray/proton/C-ions IR (*n* = 4).

**Figure 6 ijms-23-11464-f006:**
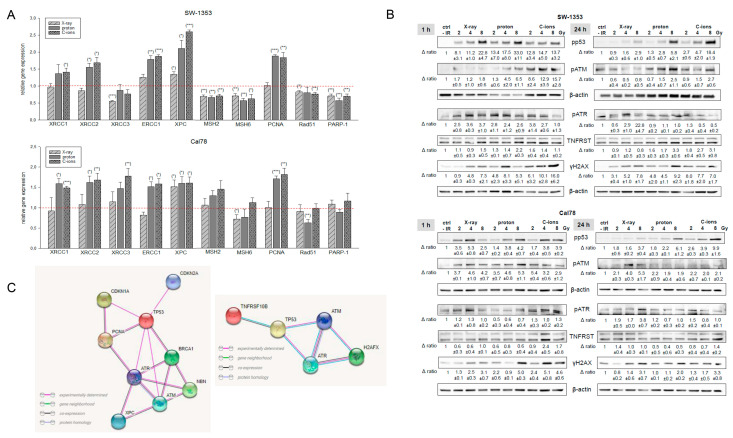
DNA repair key players. (**A**) Relative gene expression of XRCC1/2/3, ERCC1, XPC, MSH2/6, PCNA, Rad51, and PARP-1 24 h after treatment with 4 Gy X-ray (light grey striped), proton (dark gray), and C-ions (dark grey dotted) IR in SW-1353 and Cal78 chondrosarcoma cells (mean ± SD; *n* = 4; measured in triplicates). Non-IR cells were used as controls, which is represented by the red dotted line (ratio = 1). Statistical significances are defined as follows: * *p* < 0.05; ** *p* < 0.01; *** *p* < 0.001. (**B**) Protein phosphorylation pattern of SW-1353 and Cal78 chondrosarcoma cells. The influence of IR on protein phosphorylation of p53, ATM, ATR, the TNF receptor TNFRST10B, and the DNA damage marker γH2AX was evaluated by immunoblotting under non-IR control conditions (ctrl) and 1 and 24 h after 2, 4, and 8 Gy X-ray/proton/C-ion IR. β-actin was used as loading control. Δ ratio, fold change normalized to non-IR controls (mean ± SD of *n* = 3). (**C**) The STRING database interaction network revealed the relationship of the most relevant cell cycle and DNA repair gene.

**Table 1 ijms-23-11464-t001:** Cell cycle distribution of chondrosarcoma cells 24 h after 4 and 8 Gy photon (X-ray)/proton/C-ions IR (*n* = 4; mean ± SD; ** *p* < 0.01; *** *p* < 0.001; n.s.: not significant).

	SW-1353			Cal78		
	G_1_/G_0_	S	G_2_/M	G_1_/G_0_	S	G_2_/M
**ctrl 0 Gy**	56.9 ± 2.5	29.4 ± 1.1	13.7 ± 3.4	60.6 ± 2.6	27.3 ± 3.2	12.1 ± 1.2
**X-ray 4 Gy**	69.9 ± 3.6 **	14.8 ± 3.1 ***	15.2 ± 0.8 n.s.	42.7 ± 6.5 **	23.6 ± 2.6 n.s.	33.6 ± 4.1 ***
**X-ray 8 Gy**	59.1 ± 2.8	7.7 ± 1.1 ***	33.2 ± 2.3 ***	16.3 ± 1.8 ***	29.6 ± 2.4 n.s.	54.1 ± 1.5 ***
**proton 4 Gy** LET 2.9 keV/μm	63.4 ± 1.1 **	18.8 ± 0.6 ***	17.5 ± 1.0 n.s.	39.9 ± 0.8 ***	25.1 ± 0.7 n.s.	35.3 ± 1.0 ***
**proton 8 Gy** LET 2.9 keV/μm	58.1 ± 2.5 n.s.	8.9 ± 1.3 ***	33 ± 1.2 ***	17.1 ± 0.9 ***	29.8 ± 1.5 n.s.	52.6 ± 0.9 ***
**C-ions 4 Gy** LET 55.2 keV/μm	53.3 ± 4.0	7.8 ± 1.4 ***	39.5 ± 4.0 ***	19.8 ± 1.5 ***	31.2 ± 2.9	48.9 ± 1.5 ***
**C-ions 8 Gy** LET 55.2 keV/μm	28.7 ± 2.3 ***	3.6 ± 0.3 ***	67.7 ± 2.6 ***	11.8 ± 0.8 ***	36.7 ± 1.6 **	51.5 ± 1.3 ***

## Data Availability

Not applicable.

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
