# Peer review of "Cellular and Molecular Biological Alterations after Photon, Proton, and Carbon Ions Irradiation in Human Chondrosarcoma Cells Linked with High-Quality Physics Data"

_ijms, 2022, doi:10.3390/ijms231911464_

Round 1

Reviewer 1 Report

Thank you for the opportunity to review the manuscript entitled, "Cellular and molecular biological alterations after photon, proton, and carbon ions irradiation in human chondrosarcoma cells linked with high quality physics data." The authors develop a water phantom-based setup for cell culture experiments. They also analyzed human chondrosarcoma cell lines regarding viability, cell proliferation, cell cycle, and DNA repair behavior after irradiation with X-ray, proton, and carbon ions. The authors discovered a prominent G2/M arrest and showed that increasing ionization density inhibited cell viability and proliferation. The topic is essential, as chondrosarcoma is the second most diagnosed primary malignant bone tumor, and a better understanding of therapy would benefit patients. However, I have a couple comments to improve the quality of the manuscript.

  1. It would benefit the paper if the authors added a couple of sentences or an additional paragraph in the introduction where they describe potentially similar studies and what their shortcomings are. 
  2. Could the authors provide citations for the sentences in lines 221-224?
  3. My biggest concern is the lack of a dedicated limitations section. Could the authors dedicate a section for this?

Author Response

Reviewer 1 (IJMS-1908195)

Thank you for the opportunity to review the manuscript entitled, "Cellular and molecular biological alterations after photon, proton, and carbon ions irradiation in human chondrosarcoma cells linked with high quality physics data." The authors develop a water phantom-based setup for cell culture experiments. They also analyzed human chondrosarcoma cell lines regarding viability, cell proliferation, cell cycle, and DNA repair behavior after irradiation with X-ray, proton, and carbon ions. The authors discovered a prominent G2/M arrest and showed that increasing ionization density inhibited cell viability and proliferation. The topic is essential, as chondrosarcoma is the second most diagnosed primary malignant bone tumor, and a better understanding of therapy would benefit patients. However, I have a couple comments to improve the quality of the manuscript.

Authors reply: Thank you for the positive feedback on our research. All comment have been carefully considered and were taken into account as explained in the detailed point-by-point reply below.

  1. It would benefit the paper if the authors added a couple of sentences or an additional paragraph in the introduction where they describe potentially similar studies and what their shortcomings are.

Authors reply: We have added a respective but short sentence at the end of section introduction, but would like to draw the attention of reviewer 1 to the 3rd paragraph of section discussion, where additional comments were already given in the original version of the manuscript. The authors prefer to keep these sentences in section discussion, where they but our study and results, respectively, in a more specific research context than in section introduction.

  1. Could the authors provide citations for the sentences in lines 221-224?

Authors reply: We have added the following citation to this section, i.e. the publication by Park and Kang (2011) which provides an in-depth explanation about the underlying physics of particle therapy for the interested reader.

New Ref#12: Park SH; Kang JO, Basics of particle therapy I: physics. Radiat Oncol J. 2011, 29(3), 135-46.

  1. My biggest concern is the lack of a dedicated limitations section. Could the authors dedicate a section for this?

Authors reply: We have added a Limitations section at the end of the manuscript, which addresses possible uncertainties from the Radiation Physics and Radiobiology point of view, see pages 14 and 15.

Reviewer 2 Report

Review comments of ijms-1908195 manuscript

Current manuscript is entitled “Cellular and molecular biological alterations after photon, proton, and carbon ions irradiation in human chondrosarcoma cells linked with high quality physics data”. Authors have reported cancer radiotherapy with a novel water phantom-based setup for the human chondrosarcoma cancer cell line. The manuscript was demonstrated with adequate experimental results and well written with scientific sounds. Therefore, I would strongly recommend this manuscript for publication in the MDPI-IJMS journal without any further corrections.

Author Response

Reviewer 2 (IJMS- 1908195)

Current manuscript is entitled “Cellular and molecular biological alterations after photon, proton, and carbon ions irradiation in human chondrosarcoma cells linked with high quality physics data”. Authors have reported cancer radiotherapy with a novel water phantom-based setup for the human chondrosarcoma cancer cell line. The manuscript was demonstrated with adequate experimental results and well written with scientific sounds. Therefore, I would strongly recommend this manuscript for publication in the MDPI-IJMS journal without any further corrections.

Authors reply: We are very pleased with the positive review of our work and thank Reviewer 2 for her/his time and efforts.